# Benchmark and Boosted Segmentation on Dental Panoramic Radiographs

**Kaiwei Sun** [1]                                        KAIWEIS2@ILLINOIS.EDU
**Yang Feng** [2]                                        FENGYANG@ANGELALIGN.COM
**Zuozhu Liu**[*1]                                        ZUOZHULIU@INTL.ZJU.EDU.CN
[1] *Zhejiang University, China*
[2] *Angelalign Inc., China*

**Editors:** Accepted for publication at MIDL 2023

## Abstract

Panoramic radiographs, also known as orthopantomograms, are commonly utilized by dentists to gain a comprehensive understanding of the patient's oral health and perform orthodontic procedures. However, due to physician burnout and time constraints, many dentists may use them hastily which could result in medical negligence. To streamline the workflow for dentists, we establish a mission to segment five oral structures on panoramic radiographs, namely Alveolarcrest, Condyle, Neuraltube, Sinusmaxillaris and Teeth. A Cascaded Multi-scale Mask2former(CMMask2former) method is proposed for this task. For small objects, we design a multi-scale masked attention specifically for the mask area. The entire structure is designed in a two-stage cascade for localization and prediction. Our results demonstrate superior predictive performance compared to other methods.

**Keywords:** Panoramic radiographs, Orthodontic, Segmentation, Mask2former.

## 1. Introduction

Physician burnout is a significant concern that affects all dentists due to the overwhelming patient data for dentists to review and complex workflows.(Yates, 2020) Panoramic radiographs, as a way to visualize patients' oral condition; most dentists' use tends to be imprecise. The quality of care is often sub-optimal, and some preventable medical errors may happen. Meanwhile, patients are also dissatisfied with limited interactions and attention from doctors during their short clinical visits. Segmentation models for the analysis of Panoramic radiographs data is a challenging task that helps to uncover different oral structures which can reduce the workflow of dentists and give both dentists and patients a clearer visualization of their oral condition for orthodontic treatment plans.(Wu et al., 2018) In this context, we propose a mission to segment on the panoramic radiographs for five oral structures: Alveolarcrest, Condyle, Neuraltube, Sinusmaxillaris, and Teeth. Considering the Independence between teeth and overlap area with other structures, an instance segmentation is applied for the teeth class.(Lee et al., 2020) For the other four structures, we detect them with a semantic segmentation way.(Koch et al., 2019) Recently, a new deep learning paradigm that is compatible with both two segmentation methods was introduced: Masked-attention Mask Transformer (Mask2former)(Cheng et al., 2022), which could be

---

* Corresponding author

applied to this mission. It is a transformer-based encoder-decoder framework for Universal Image Segmentation. The semantic segmentation for Alveolarcrest, Condyle, Neuraltube, and Sinusmaxillaris is still a challenging task, especially for Neuraltube which is slender. To enhance the performance for small objects in Neuraltube, we specially designed a Cascaded structure based on Mask2former with Multi-scale Masked attention for its segmentation. The proposed framework Cascaded Multi-scale Mask2former(CMMask2former) offers state-of-the-art performance in the segmentation of panoramic radiographs.

## 2. Methods

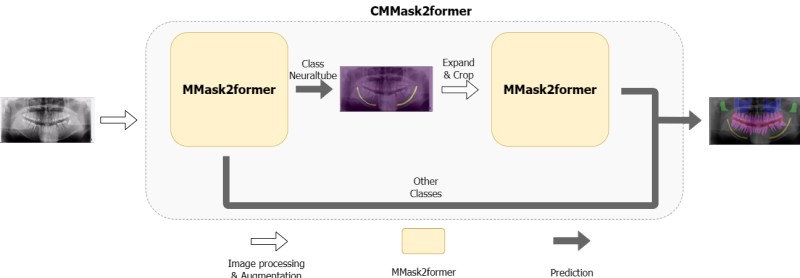

Figure 1: CMMask2former Framework: Two stages of Multi-scale Masked attention Mask-former (MMask2former) in a cascaded way.

The schematic overview of our CMMask2former framework is shown in Figure 1. Our framework is constructed based on Mask2former(Cheng et al., 2022). For the slender and small characteristics of Neuraltube, it may be challenging to learn the features of the object and find the boundary distinction, hence we first design a Multi-scale Masked attention to localize the object region and import more surrounding information of the object. Based on the Masked attention(Cheng et al., 2022), we adjust the masked attention map to the bounding box of predicted area and use all of the previous masked attention maps to generate the attention map for the next layer. Our Multi-scale masked attention modulates the attention matrix via:

$$X_l = softmax(\frac{1}{l}\sum_{i=0}^{l-1} BOX(M_i) + Q_l K_l^T)V_l + X_{l-1}. \tag{1}$$

Here, l is the layer index, $X_l \in R^{NC}$ refers to NC-dimensional query features at the lth layer and $Q_l$. $Q_l \in R^{NC}$ is the query feature. $K_l, V_l \in R^{H_l W_l C}$ are the image features respectively, and $H_l$ and $W_l$ are the spatial resolution of image features. $M_{l-1} \in \{0,1\}^{NH_l W_l}$ is the binarized prediction output (thresholded at 0.5) from the previous layer. BOX is the function to fill the prediction of binary mask $M_i$ to a shape of its bounding box area.

Overall, we employ a cascaded structure consisting of two Mask2formers with Multi-scale masked attention. The initial MMask2former is utilized for object region localization. Following a 10% expansion of its width and height, with the center fixed, we extract the

predicted region from the original image. The resulting cropped image is then forwarded to the subsequent MMask2former for further prediction

## 3. Experiments and conclusion

**Data Set** We analyze Panoramic radiographs from 466 patients who received orthodontics service from Angelalign. The dataset is randomly separated into three data sets for training, validation, and testing with a ratio of 7:1:2. All patients' Panoramic radiographs are manually annotated with five classes including Alveolarcrest, Condyle, Neuraltube, Sinusmaxillaris, and Teeth (each tooth is independent).

**Semantic segmentation** We evaluate the performance of CMMask2former against state-of-the-art models on our dataset, as presented in Table 1. The comparation models include Unet++(Zhou et al., 2020), MedT(Valanarasu et al., 2021), UCTransNet(Wang et al., 2022) and Mask2former(Cheng et al., 2022). Additionally, we present the results of our method without Multi-scale masked attention(MMA) or Cascaded structure(CS) over CM-Mask2former to demonstrate the effectiveness of our modules.

Table 1: Semantic segmentation on our data set with four classes

| Model | $IoU_{Alveolarcrest}$ | $IoU_{Condyle}$ | $IoU_{Neuraltube}$ | $IoU_{Sinusmaxillaris}$ | $mIoU$ |
|---|---|---|---|---|---|
| Unet++ | 72.731 | 29.732 | 17.036 | 48.754 | 42.063 |
| MedT | 88.089 | 72.885 | 35.316 | 79.493 | 68.946 |
| UCTransNet | 91.472 | 83.236 | 59.666 | 85.601 | 79.994 |
| Mask2former | 92.566 | 89.276 | 62.457 | 87.575 | 82.968 |
| Ours(w/o CS) | 92.840 | 88.071 | 64.863 | 88.021 | 83.449 |
| Ours(w/o MMA) | 92.902 | 89.413 | 66.144 | 88.047 | 84.127 |
| Ours | 92.654 | 89.902 | 67.537 | 87.589 | 84.421 |

**Instance segmentation** We compare CMMask2former with Mask R-CNN(He et al., 2018) on our data set in Table 2. For this part, we do not import our modules, hence, the results of Mask2former and our model are close.

Table 2: Instance segmentation on our data set with teeth class

| Model | $AP_{Teeth}$ |
|---|---|
| Mask R-CNN | 52.034 |
| Mask2former | 74.345 |
| Ours | 74.381 |

**Conclusion** In conclusion, this paper provides a preliminary insight into the segmentation task of five oral structures on Panoramic radiographs. We propose a novel Cascaded Multi-scale Mask2former (CMMask2former) method for this challenging segmentation task. Experimental results on our dataset demonstrate the effectiveness of our proposed modules.

**Acknowledgments**
This work is supported by the National Natural Science Foundation of China (Grant No. 62106222), the Natural Science Foundation of Zhejiang Province, China(Grant No. LZ23F020008) and the Zhejiang University-Angelalign Inc. R&D Center for Intelligent Healthcare.

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
