# OpenReview forum: "Benchmark and Boosted Segmentation on Dental Panoramic Radiographs"
_MIDL.io/2023/Short_Paper_Track — MIDL 2023 Short paper track Poster_

### Official Review · Reviewer_epCP · 2023-04-23
**Well-motivated but heavily builds on existing work**

**Rating:** 5
**Confidence:** 5

**Review:**

The paper proposes a cascaded network based on cascading two Mask2formers to segment five oral structures on Panoramic radiographs. Experiments show marginal improvements compared to Mask2former except for the neuraltube class, which is a small thin structure. While well-motivated, the paper heavily builds on existing work.

---

### Official Review · Reviewer_hJmm · 2023-04-24
**Review of: Benchmark and Boosted Segmentation on Dental Panoramic Radiographs**

**Rating:** 7
**Confidence:** 4

**Review:**

This abstract introduces a variation of a mask transformer (Cheng et al 2022) for a set of segmentation tasks in dental radiographs. It includes a demonstration on a small dataset.

Overall I would focus on explaining the reasoning for your method. I was able to extract it from section 2:
1) the Neural Tube is a narrow, hard to segment region.
2) thus, we provide our fixes, which gives bounding box context to the attention method
3) To show the usefulness of point 2, we should look specifically at the Neural Tube segmentation results and the improvement there.
This message is in your current paper, but needs to be apparent immediately. I think it's a strong (if simple) story, and can motivate why you make the changes to the mask transformer.

I also strongly recommend larger and more visual examples.

Minor comment:
define a mission -> define five segmentation tasks for oral radiographs: ...

I recommend this paper to be accepted, I think it would make a good poster presentation and meets acceptance criteria for short papers.